# Application of Absolute Nodal Coordinate Formulation in Calculation of Space Elevator System

**Shihao Luo** [1] , **Youhua Fan** [1,*] **and Naigang Cui** [2]

1 School of Science, Harbin Institute of Technology, Shenzhen 518055, China; luoshihao@stu.hit.edu.cn
2 School of Astronautics, Harbin Institute of Technology, Harbin 150001, China; cuinaigang1965@hit.edu.cn
* Correspondence: yhfan@hit.edu.cn

**Abstract:** The space elevator system is a space tether system used to solve low-cost space transportation. Its high efficiency, large load, reusability and other characteristics have broad application prospects in the aerospace field. Most of the existing mechanical models are based on "chain-bar" and a lumped mass tether model, which cannot effectively reflect the flexible behaviour of the rope of space elevator system. To establish an accurate mechanical model, the gradient deficient beam elements of the absolute nodal coordinate formulation (ANCF) are used to build the mechanical model of the space elevator system. The universal gravitation and centrifugal force in the model are derived. The calculation results of the ANCF model are compared with the results of the finite element method (FEM) and lumped mass (LM) models. The results show that the calculation results of the ANCF method are not very different from the results of the FEM and LM models in the case of axial loading. In the case of lateral loading, the calculation results of the ANCF method are basically the same as the results of the FEM and LM models, but can better reflect the local flexible deformation of the space elevator rope, and have a better calculation stability than FEM. Under the same calculation accuracy, the ANCF method can use fewer elements, and the speed of convergence is faster than the FEM and LM models.

**Keywords:** space elevator; absolute nodal coordinate formulation; lateral loading

## 1. Introduction

As a potential new space transportation system for transporting the payload to outer space, the main structure of the space elevator system mainly consists of four parts: ground anchor point, rope, climber and zenith anchor. The whole system relies on the centrifugal force generated by the rotation of the earth to keep the system stable and transport the payload through the climber.

The concept originated from the invention of Tsiolkovsky, and was cemented in engineering terms by Pearson [1]. It was not possible to obtain the desired strength of the tether material until the discovery of carbon nanotubes [2]. At the end of the 20th century, NASA established a research group led by Dr. Edward and demonstrated the feasibility of the space elevator system [3]. To fully understand the potential commercial application of space elevator, the International Space Elevator Association (ISEC) introduced the concept of Galaxy port [4]. Through the life cycle assessment of the space elevator, Harris [5] found that the design of the space elevator is an environmentally and economically sustainable choice for rail transportation. Shi [6] studied the dynamics of a partial space elevator system with multiple climbers and applied the optimal control to develop optimal operation modes to suppress the liberation of the partial space elevator.

Pearson [1] established the continuous static model of a rope element based on differential equations, and first deduced the gradual function model of the rope's cross-sectional area. However, the elasticity of the rope was not considered, so Aravind [7] and Cohen [8] established the system static equilibrium equation considering the rope elasticity on the

basis of Pearson, deduced the cross-sectional area gradient function model of the equatorial space elevator elastic rope, and conducted relevant research on the system parameter of the gradient section space elevator system, which laid the foundation for the subsequent research on the dynamics of the space elevator system.

The rigid rod tether model has fewer degrees of freedom and simple analysis. Wang [9] studied the stability characteristics of the space elevator system near the equilibrium point based on the small angle hypothesis through the rigid rod tether model. The chain rod model discretizes the continuous rope into a system composed of finite rigid rods or elastic rods [10]. Woo [11] used the chain rod model to study the stress of some space elevator systems, and explored the influence of climbers on the vibration law of some space elevator systems. The lumped mass tether model discretizes the rope model into a series of mass points connected by spring dampers, simulates the axial stiffness of the rope through the elasticity of the spring, and reflects the flexibility of the rope with the rotation angle of the adjacent spring. Based on the lumped mass tether model, Williams [12] studied the vibration influence of the climber on the space elevator system, proposed the relevant control strategy, and carried out the modal analysis of the system. Wang [13] carried out a dynamic modeling of deployment latitude based on the "chain-bar" tether model and lumped mass tether model, respectively, and studied the deployable latitude range of non-equatorial space elevator system.

Neither the "chain-bar" nor the lumped mass tether model can truly reflect the constitutive mechanical characteristics of the flexible rope, and cannot accurately describe the large deformation movement of the flexible rope. A high number of elements is necessary to ensure the accuracy of the model; thus, the simulation calculation is inefficient. Shabana [14] proposed the absolute nodal coordinate formulation (ANCF), which is based on the theory of continuum mechanics and mainly solves the dynamics of large deformation flexible bodies. The ANCF uses position vector and derivative to describe the element nodes, which can avoid the limitation of the small rotation angle in the traditional finite element method, and describe arbitrary translation, rotation and deformation [15]. The mass matrix in ANCF is constant, and there are no centrifugal force terms and Coriolis force terms in the dynamic equation, which further simplifies the numerical calculation [16]. ANCF was employed to establish the dynamics model of space tether in new applications due to its ability to describe the nonlinearity and large deformation of the tether [17], including the effects of solar wind on an electric sail (E-sail) [18], deployment dynamics and debris capture of tethered space net [19], and tethered spacecraft formation [20]. The ANCF method has also been applied to the dynamics of tethered satellites and partial space elevators [21–23].

It is necessary to accurately obtain the space elevator system's mechanical properties for its design and construction. The mechanical model established for analysis needs to be as close to the actual situation as possible. Most of the existing mechanical models are based on the "chain-bar" and lumped mass tether model, which cannot effectively reflect the flexible behavior of the rope of space elevator system. Thus, in this paper, the ANCF method will be applied to the calculation of the space elevator system, in order to describe the rope flexibility of the space elevator system.

## 2. Tether Modelling

The typical space elevator system model comprises a tether, climbers and a zenith anchor. It can be considered that the anchor point of the tether is at a fixed location while the system oscillation has little impact on the floating platform. The centrifugal force would pull the tether perpendicular to the Earth's axis while the tether would also droop toward the equator owing to gravity, which points to the center of the Earth. The space elevator system model is depicted in Figure 1. The center of the Earth $O$ is assumed to be inertially fixed and is used as the origin of the inertial frame. $O_0$ is the anchor point of the tether on the earth's equator. The $X$ axis points from point $O$ to point $O_0$ and is perpendicular to the ground. The $Z$ axis points to the North Pole along the Earth's rotation axis. The $Y$ axis is

perpendicular to the $X$ axis and $Z$ axis, $\omega$ is the angular velocity of the system is equal to the rotation rate of the Earth. $L_T$ is the nominal (unstressed) length of the tether. $m_c$ is the mass of the counterweight at the end of the tether. $\theta$ is the angle between the projection of the rope of the space elevator system in the $XY$ plane and the $X$ axis. $\phi$ is the angle between the projection of the rope of the space elevator system in the $XY$ plane and itself.

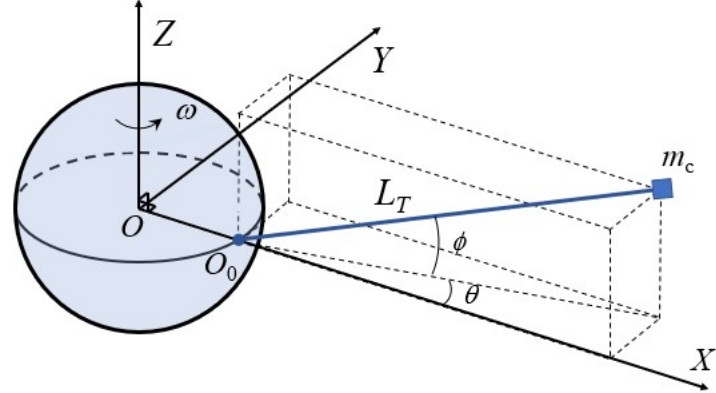

**Figure 1.** Space elevator system model.

The rope of the space elevator system is a kind of slender flexible body, in which the scale of one dimension (length) is much larger than the other two dimensions (section), so it can be modeled based on beam element. ANCF beam elements can generally be divided into two categories. The first one is the beam element without considering the transverse shear effect. This kind of element describes the displacement and deformation field of the beam through the node position coordinates and their longitudinal derivatives. Due to the lack of transverse derivatives, it is usually called the gradient default beam element. The other one is the beam element considering the transverse shear effect. This kind of element introduces the transverse derivative of the node; thus, the section can be deformed. The section size of the rope of the space elevator system is much smaller than the length, so the shear effect can be ignored. In this paper, a flexible rope model is established based on the rope beam element with gradient default beam element.

The model of ANCF rope element is shown in Figure 2. $O - XYZ$ is the absolute coordinate system of space. $L$ is the length of the rope element. $x$ is the coordinate of the element in the length direction. $\vec{q_1}$ and $\vec{q_2}$ are the generalized coordinates of the two nodes of the rope element. $\vec{r}(x)$ is the absolute coordinate of the point on the element, whose element coordinate is $x$.

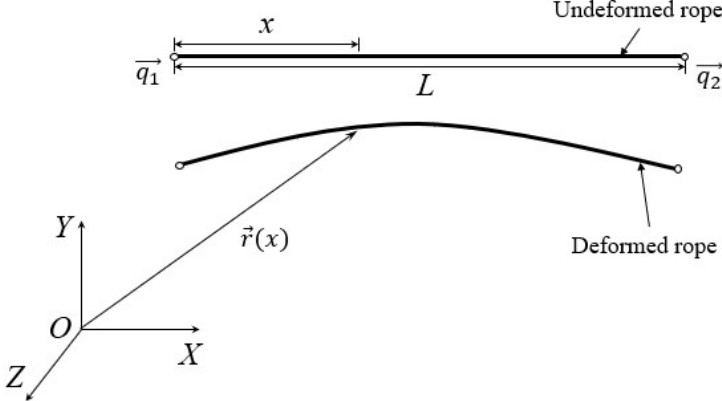

**Figure 2.** Model of ANCF rope element.

The node coordinates of the gradient default rope beam element are composed of the node position and its derivative to the axial element coordinates. For a three-dimensional two node element with length $L$, the node coordinates can be expressed as:

$$\vec{q}_e = \begin{bmatrix} \vec{q}_1 & \vec{q}_2 \end{bmatrix}^\mathsf{T} = \begin{bmatrix} \vec{r}^\mathsf{T}(x=0) & \vec{r}_x^\mathsf{T}(x=0) & \vec{r}^\mathsf{T}(x=L) & \vec{r}_x^\mathsf{T}(x=L) \end{bmatrix}^\mathsf{T} \tag{1}$$

The displacement field function of the element can be obtained by cubic Hermite interpolation of the node coordinates of the element:

$$\vec{r}(x,t) = \mathbf{S}(x)\vec{q}_e(t) \tag{2}$$

where $\mathbf{S}(x)$ is the interpolation basis function, that is, the element shape function, expressed as follows:

$$\mathbf{S}(x) = \begin{bmatrix} S_1(x)\mathbf{I_3} & S_2(x)\mathbf{I_3} & S_3(x)\mathbf{I_3} & S_4(x)\mathbf{I_3} \end{bmatrix} \tag{3}$$

where $\mathbf{I_3}$ is third order identity matrix, and $S_i(x)$ can be presented as:

$$\begin{aligned} S_1(x) &= 1 - 3\xi^2 + 2\xi^3 \\ S_2(x) &= L(\xi - 2\xi^2 + \xi^3) \\ S_3(x) &= 3\xi^2 - 2\xi^3 \\ S_4(x) &= L(-\xi^2 + \xi^3) \end{aligned} \tag{4}$$

where $\xi = x/L$.

As shown in Equation (2), the shape function and node coordinates only depend on the element coordinates and time variables, respectively, so the element displacement derivative, velocity and acceleration are expressed as follows:

$$\begin{aligned} \vec{r}_x(x,t) &= \mathbf{S}_x(x)\vec{q}_e(t) \\ \vec{r}_{xx}(x,t) &= \mathbf{S}_{xx}(x)\vec{q}_e(t) \\ \dot{\vec{r}}(x,t) &= \mathbf{S}(x)\dot{\vec{q}}_e(t) \\ \ddot{\vec{r}}(x,t) &= \mathbf{S}(x)\ddot{\vec{q}}_e(t) \end{aligned} \tag{5}$$

For a rope element with cross-sectional area $A$ and density $\rho$, its kinetic energy is:

$$T^e = \frac{1}{2}\int_0^L \rho \int_A \dot{\vec{r}}^\mathsf{T}\dot{\vec{r}}\,\mathrm{d}A\mathrm{d}x = \frac{1}{2}\dot{\vec{q}}_e^\mathsf{T}\mathbf{M}^e\dot{\vec{q}}_e \tag{6}$$

where $M^e$ is the element mass matrix, which can be presented as:

$$\mathbf{M}^e = \int_0^L \rho A S(x)^\mathsf{T} S(x)\mathrm{d}x \tag{7}$$

It can be seen from the displacement field function of the element that the axial and bending deformation are considered in the element. The axial strain $\varepsilon$ and curvature $\kappa$ can be presented as:

$$\varepsilon = \frac{1}{2}(\vec{r}_x^\mathsf{T}\vec{r}_x - 1) \tag{8}$$

$$\kappa = \frac{\|\vec{r}_x \times \vec{r}_{xx}\|}{\|\vec{r}_x\|^3} \tag{9}$$

According to Euler Bernoulli beam theory, the element strain energy can be presented as:

$$U_e^e = \frac{1}{2}\int_0^L (EA\varepsilon^2)\mathrm{d}x + \frac{1}{2}\int_0^L (EJ\kappa^2)\mathrm{d}x \tag{10}$$

where $E$ and $J$ are the Young's modulus and the section moment of inertia of the element material.

The generalized elastic force of the element can be obtained by calculating the partial derivative of the strain energy in Equation (10) with respect to the generalized coordinate:

$$\vec{Q}_e^e = \int_0^L EA\varepsilon \frac{\partial \varepsilon}{\partial \vec{q}} dx + \int_0^L EJ\kappa \frac{\partial \kappa}{\partial \vec{q}} dx \tag{11}$$

The generalized structural force is composed of universal gravitation and centrifugal force. As shown in Figure 1, there is no difference between the perturbation of universal gravitation in the $XY$ plane or $XZ$ plane, but the centrifugal force is different in the $XY$ plane and the $XZ$ plane. This is because the calculation of universal gravitation depends on the distance between the particle and the earth's center, while centrifugal force depends on the distance between the particle and the earth's rotation axis. As a result, the $z$ component in $\vec{r}$ has no contribution to centrifugal force, a transformation matrix $\mathbf{C}$ is set to calculate centrifugal force, which can be presented as:

$$\mathbf{C} = \begin{pmatrix} 1 & 0 & 0 \\ 0 & 1 & 0 \\ 0 & 0 & 0 \end{pmatrix} \tag{12}$$

The gravitational force $\vec{Q}_g^x$ and centrifugal force $\vec{Q}_c^x$ with the element coordinate $x$ can be expressed as:

$$\vec{Q}_g^x = -\frac{\mu m_x^e}{\vec{r}(x)^2} \tag{13}$$

$$\vec{Q}_c^x = m_x^e \omega^2 \mathbf{C}\vec{r}(x) \tag{14}$$

where $\mu$ is the gravitational constant, and the point mass $m_x^e = \rho A$.

For objects with mass $m_o$ in the geostationary orbit, the gravitational force is equal to the centrifugal force, and we can obtain:

$$\frac{\mu m_o}{R_g^2} = m_o \omega^2 R_g \tag{15}$$

where $R_g$ is the geosynchronous orbit radius.

$\omega^2$ can be taken from Equation (15) and integrate Equation (13), Equation (14) along the length of the element to obtain:

$$\vec{Q}_g^e = -\int_0^L \frac{\mu \rho A}{\vec{r}(x)^2} dx \tag{16}$$

$$\vec{Q}_c^e = \int_0^L \frac{\mu \rho A}{R_g^3} \mathbf{C}\vec{r}(x) dx \tag{17}$$

Then, for an element with constant density $\rho$ and a cross-sectional area $A$, the generalized structural force $\vec{Q}_s^e$ can be shown as:

$$\vec{Q}_s^e = \vec{Q}_g^e + \vec{Q}_c^e = \mu \rho A \int_0^L \left( \frac{\mathbf{C}\vec{r}(x)}{R_g^3} - \frac{1}{\vec{r}(x)^2} \right) dx \tag{18}$$

The external forces acting on the rope mainly include concentrated load and distributed load. The generalized force of the concentrated load $\vec{P}$ acting on $x_p$ can be presented as:

$$\vec{Q}_{e1}^e = \mathbf{S}(x_p)^\mathsf{T} \vec{P} \tag{19}$$

The generalized force of distributed load $\vec{F}$ can be presented as:

$$\vec{Q}_{e2}^e = \int_0^L \mathbf{S}(x)^\mathsf{T} \vec{F} \mathrm{d}x \tag{20}$$

The system mass matrix $\mathbf{M}$ can be assembled from the mass matrix of each element using the same method as the finite element method. The system kinetic energy is shown as:

$$T = \vec{q}^\mathsf{T} \mathbf{M} \vec{q} \tag{21}$$

where $\vec{q}$ is the generalized coordinate vector composed of the coordinates $\vec{q}_e$ of each node.

The assemblies of the generalized force $Q_e$, $Q_s$, $Q_{e1}$ and $Q_{e2}$ are slightly simpler, because they are one-dimensional vectors.

As a result of the use of generalized coordinates to describe the dynamical state of rope elements, the dynamic model of steel cable capture mechanism, which is a constrained multi-body system, can be expressed by a unified system equation (differential-algebraic equation) on the basis of the Lagrange equation:

$$\frac{\mathrm{d}}{\mathrm{d}t}\left(\frac{\partial T}{\partial \dot{\vec{q}}}\right)^\mathsf{T} + \frac{\partial U_e}{\partial \vec{q}} = \vec{Q}_s + \vec{Q}_{e1} + \vec{Q}_{e2} \tag{22}$$

In the static calculation, the quantity related to the time derivative in the dynamic equation can be directly eliminated, and the dynamic equation degenerates into:

$$\vec{Q}_e = \vec{Q}_s - \vec{Q}_{e1} - \vec{Q}_{e2} \tag{23}$$

The setting of the boundary conditions is also the same as the finite element method. The initial value $\vec{q}_{e0}$ of each node is shown as:

$$\vec{q}_{e0} = \begin{bmatrix} r_x & 0 & 0 & 1 & 0 & 0 \end{bmatrix}^\mathsf{T} \tag{24}$$

where $r_x$ is the absolute coordinate of $X$-axis of the node.

Since the rope is fixed at the anchor point, the 6-degrees-of-freedom of anchor point node is set to 0, and (the number of nodes $-$ 1) $\times$ 6 equations can be obtained. Let

$$F(\vec{q}) = \vec{Q}_e(\vec{q}) - \vec{Q}_s(\vec{q}) + \vec{Q}_{e1}(\vec{q}) + \vec{Q}_{e2}(\vec{q}) = 0 \tag{25}$$

The Jacobian matrix of $F(\vec{q})$ should be obtained by numerical methods for the complicated expression of the generalized elastic force; then, the generalized coordinates $\vec{q}$ can be solved by the Newton Raphson iterative method, and the absolute coordinates of each node can be obtained by Equation (2).

### 3. Numerical Simulation

In this section, a study of rope axial stress profile of the elevator systems will be conducted, consisting only of a tether and counterweight and without any payload. A comparison of the three tether models(lumped mass tether model (LM), finite element method (FEM) and absolute nodal coordinate (ANCF)) is presented to verify the efficiency and accuracy of the above approaches.

#### 3.1. LM Model

The LM model of article [13] is shown in Figure 3.

In the LM model, the tether is discrete into n + 1 mass points and n massless springs. Each mass point is connected by two springs at the front and back to form a line. For each mass point, the force diagram is shown in Figure 4, where $m_i$, $F_{Ci}$, $F_{Gi}$ and $\mathbf{r}_i$ are the mass, centrifugal force, the gravitational force and the position vector of the $i$-th discrete mass point, respectively. $l_i$ is the length of the $i$-th spring, $T_i$ is the force of the $i$-th spring, $\alpha_i$ is

the angle between the *i*-th spring and the *X*-axis and $\theta_i$ is the angle between the direction of $F_{Gi}$ and the *X*-axis.

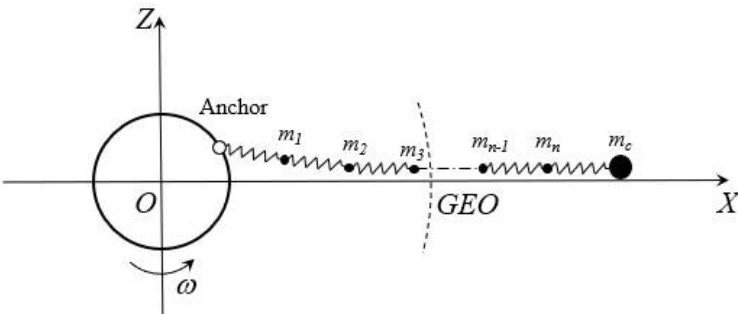

**Figure 3.** The lumped mass tether model.

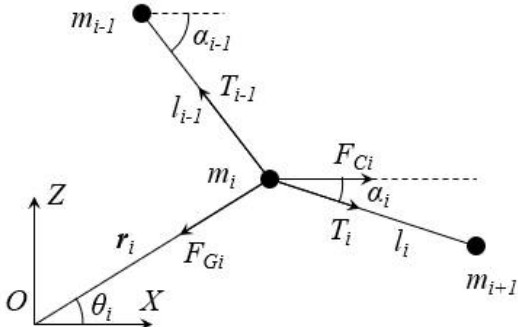

**Figure 4.** Forces acting on the lumped-mass.

According to the force balance, the balance equation of the mass point can be obtained:

$$
\begin{aligned}
0 &= T_i \cos \alpha_i - T_{i-1} \cos \alpha_{i-1} + F_{Ci} - F_{Gi} \cos \theta_i \\
0 &= -T_i \sin \alpha_i + T_{i-1} \sin \alpha_{i-1} - F_{Gi} \cos \theta_i
\end{aligned}
\tag{26}
$$

where $F_{Ci}$, $F_{Gi}$ and $\theta_i$ can easily be obtained from the position vector $\mathbf{r}_i$ of the mass point. The position vector can be obtained from the anchor point, the angle $\alpha_i$ and the length of the each spring:

$$
\mathbf{r}_{i+1} = \mathbf{r}_i + \begin{bmatrix} l_i \cos \alpha_i & l_i \sin \alpha_i \end{bmatrix}^{\mathsf{T}}
\tag{27}
$$

where $l_i = (\frac{T_i}{EA} + 1) \times l_{0i}$, $EA$ and $l_{0i}$ are the axial stiffness and the original length of the spring, respectively.

Add the boundary conditions at the end of the tether:

$$
\begin{aligned}
0 &= -T_n \cos \alpha_n + F_{Cc} - F_{Gc} \cos \theta_c \\
0 &= T_n \sin \alpha_n - F_{Gc} \cos \theta_c
\end{aligned}
\tag{28}
$$

From Equations (26) and (28), the spring force $T$ and the angle $\alpha$ can be solved by numerical methods.

### 3.2. Finite Element Method

The FEM is realized by ANSYS software; the element is BEAM188 with a rigid cross-section. Since ANSYS cannot handle the loading of acceleration with coordinate changes, some extra calculation steps should be taken. The logic diagram of the calculation is shown in Figure 5.

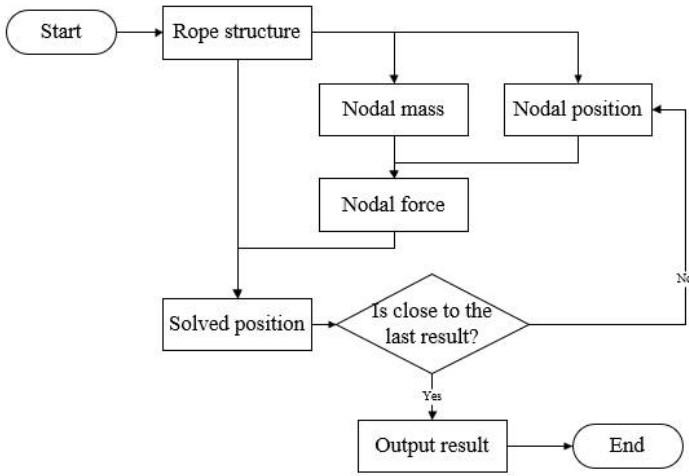

**Figure 5.** Logic diagram of FEM calculation.

The mass matrix is derived by the established structure of the rope. With this mass matrix, together with the nodal position, the gravitational and centrifugal force can be calculated. In addition, the force can be applied as a nodal force back to the structure. When the new nodal position is calculated, if it is close to the last result, the result will be output and calculation ends, otherwise the nodal position is updated and calcultation continues.

### 3.3. System Parameters

To verify the validity and precision of the methods proposed in this section, the anchor point is set on the equator. The detailed system parameters are shown in Table 1.

**Table 1.** System parameters.

| $R_e$ (m) | $R_g$ (m) | $\mu$ (N m$^2$ kg$^{-1}$) | $m_c$ (kg) | $L$ (m) | $E$ (Pa) | $\rho$ (kg m$^{-3}$) | $A$ (m$^2$) |
|---|---|---|---|---|---|---|---|
| $6.37 \times 10^6$ | $4.22 \times 10^7$ | $3.99 \times 10^{14}$ | $2.49 \times 10^9$ | $6 \times 10^7$ | $1000 \times 10^9$ | 1300 | 0.01 |

$R_e$ is the radius of the earth, $R_g$ is the geosynchronous orbit radius, $\mu$ is the geocentric gravitational constant, $m_c$ is the mass of the zenith anchor, $L$ is the total height of the space elevator system, $E$ is the Young's modulus of the rope material and $\rho$ is its density, $A$ is the area of the rope cross-section.

The element numbers of each method are shown in Table 2.

**Table 2.** Element number of each method.

| Method | Below the Geosynchronous Orbit | Above the Geosynchronous Orbit |
|---|---|---|
| LM | 1000 | 400 |
| FEM | 1000 | 400 |
| ANCF | 200 | 80 |

## 4. Results

### 4.1. Axial Loading

In Figure 6, it can be observed that the trend of the results calculated by the three methods is roughly the same. The stress results of FEM and LM are very close: the maximum stress point is around the geosynchronous orbit, and the value around 80 GPa. The result of ANCF is little smaller than the others. This is because the deformation represented by LM and FEM is Cauchy stress and true strain, which is defined by studying the force acting on an infinitesimal area element in a deformed body. Both the force component and the normal of the region have a fixed direction in space. This means that if the stressed object is subjected to pure rotation, the actual value of the stress component will change. The initial uniaxial stress state can be transformed into a total tensor containing normal stress

and shear stress components. On the other hand, ANCF use Green–Lagrange strain to represent deformation, which contains the displacement derivative relative to the original configuration. Therefore, these values represent the strain in the material direction, similar to the behavior of the second kind of Piola Kirchhoff stress. However, it must be recognized that, even for the uniaxial case, the Green–Lagrange strain is strongly nonlinear relative to the displacement. If an object is stretched to twice its original length, the Green–Lagrange strain is 1.5 in the stretching direction. If the object is compressed to half its length, the strain will be −0.375. Thus, in Figure 7, the deformation of *x*-axis of ANCF is little smaller than the other results.

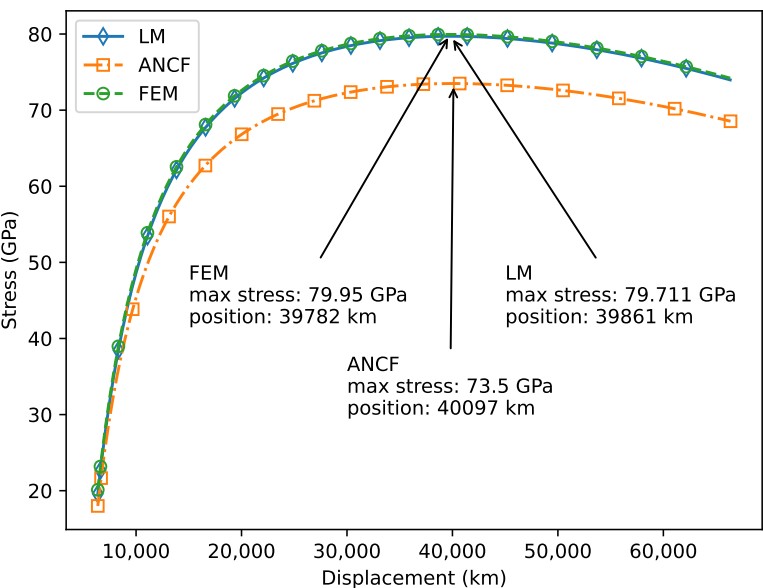

**Figure 6.** Stress results of three method.

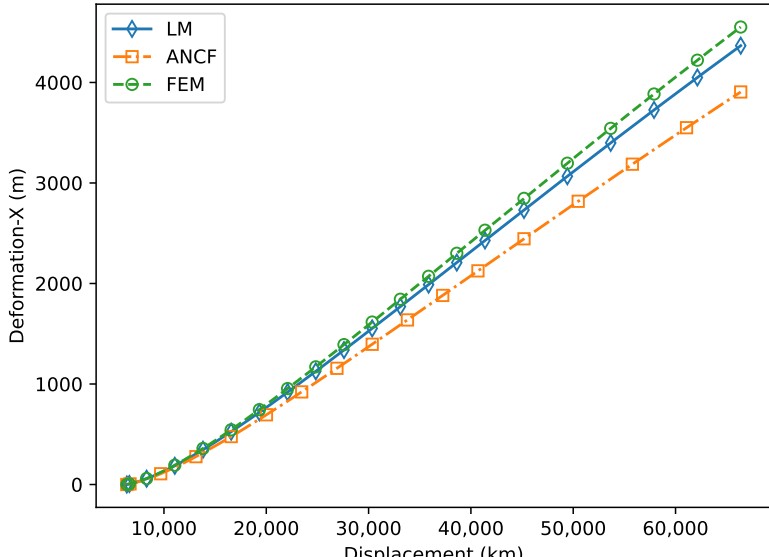

**Figure 7.** Deformation of *x* axis of three method.

### 4.2. Lateral Loading

Then, an external force is set at the top of the space elevator system in y-direction, whose value is 1000 N. The result is shown in Figures 8–10.

In Figure 8, it can be observed that deformation of *y*-axis with three method is roughly the same. The deformation of ANCF is little bigger than the others. Focusing on the

elements near the ground, as shown in Figure 9, it is expected that the contour line of the rope should resemble an S-shaped curve. The LM model cannot correctly calculate the contour of the rope as the rod element does not consider the curvature of the rod, while it can still be approximated when the number of rod elements is sufficient. The beam element of the FEM considers the curvature; however, because of the approximated curvature, the element stiffness is a constant matrix. Although the nonlinear equations can be linearized, the correlation between the element and the space coordinates is lost. The result of the ANCF method is a similar S-shaped curve. In other words, the rope angle is continuous and smooth. When the angle of the grounding point is 0, there will be no jump in the angle value.

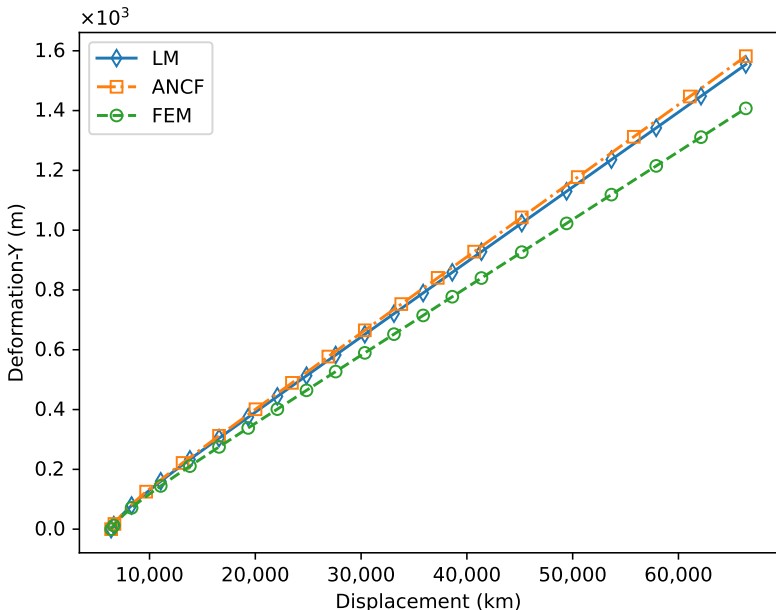

**Figure 8.** Deformation Y results of three method.

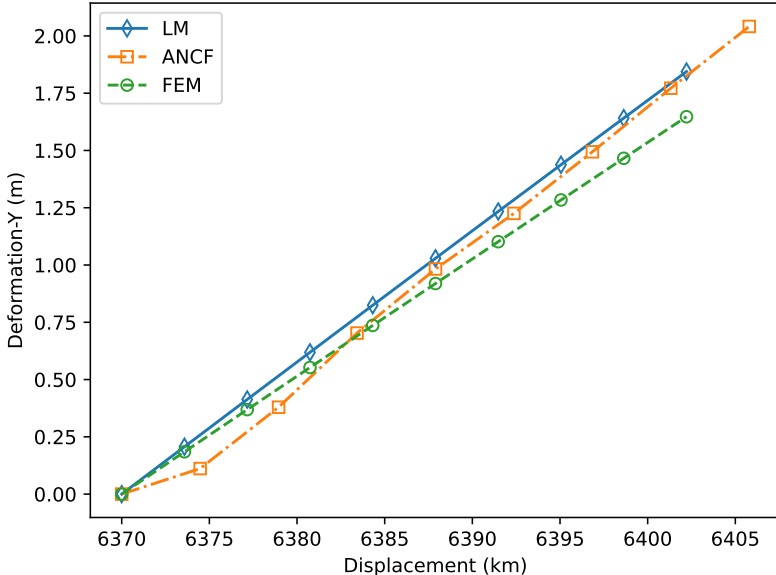

**Figure 9.** Deformation Y close to the ground.

In Figure 10, as the deflection of the element is not considered in the LM model, it can be observed that the element rotation angle of the LM model is basically kept constant. When the number of the element reaches a certain amount, the flexibility of the rope can

be approximated, but will greatly increase the cost of calculation. The element rotation angle of the FEM has an oscillating process, and finally approaches a certain value. In contrast, the element rotation angle of the ANCF method changes smoothly, and the final convergence value is slightly larger.

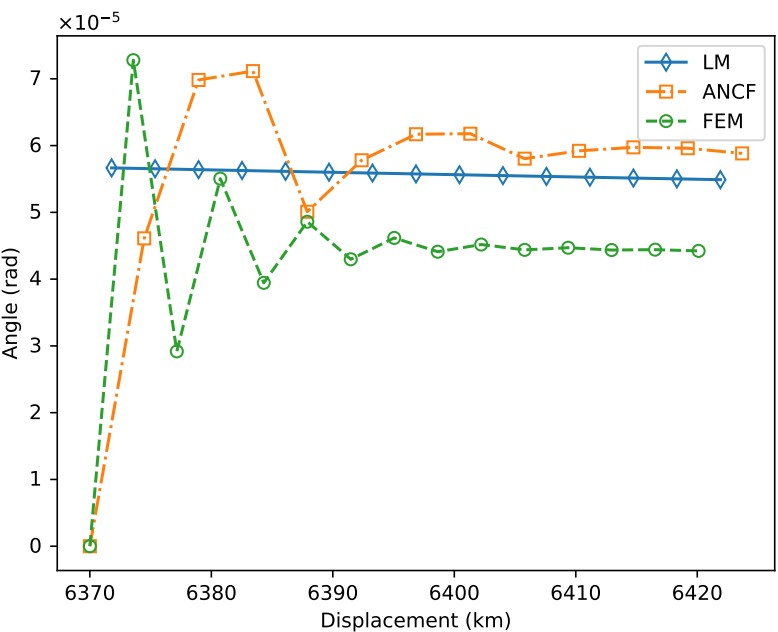

**Figure 10.** Nodal angle $\theta$ close to the ground.

Similarly, when the *Y*-axis loading turns to the *Z*-axis, the resulting deformation curve is shown in Figure 11. Since the centrifugal force has no contribution to the coordinates in the *Z* direction, the deformation of the *Z* axis is much smaller than that of the *Y* axis, and the results calculated by each method are closer.

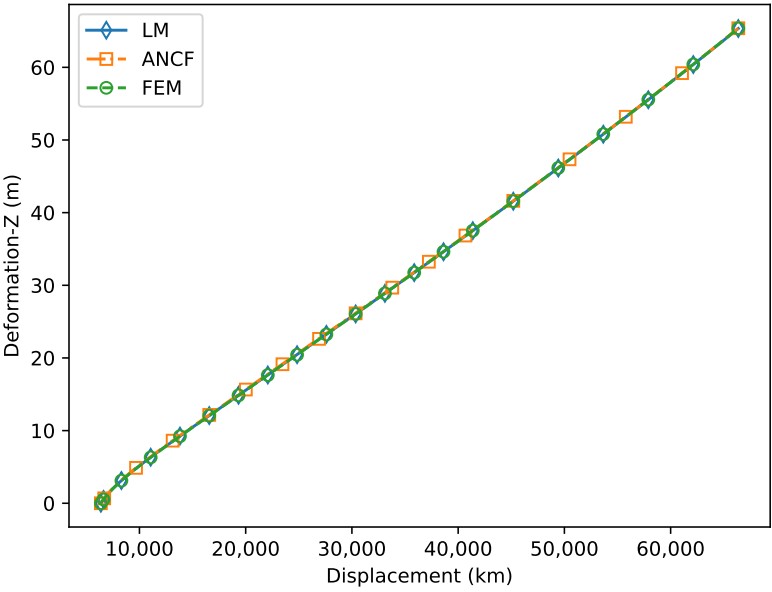

**Figure 11.** Deformation Z results of three method.

Since a section of the rope is fixed, the outline of the rope should be an S-shaped curve. It can be seen from Figure 12 that the results calculated by the ANCF method fit the S-curve better. Therefore, when studying the contour of the rope close to the ground, the calculation result of ANCF is more reasonable.

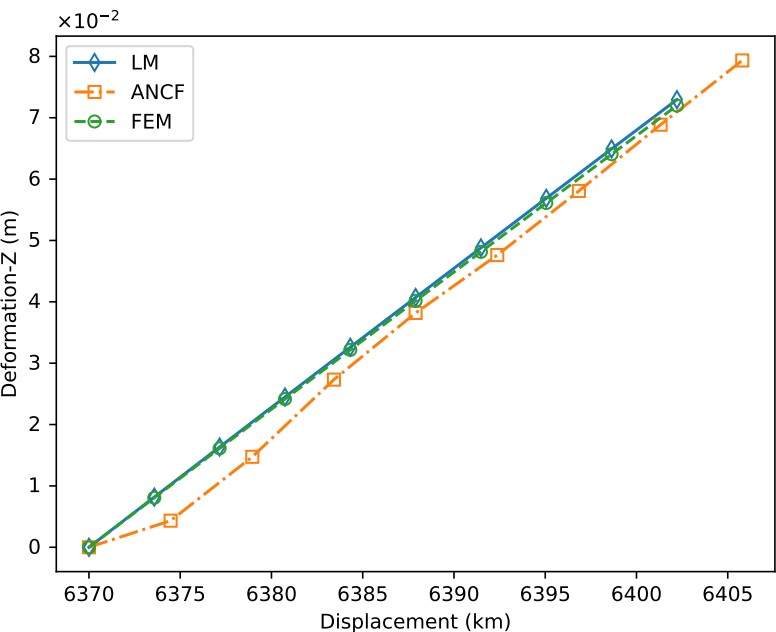

**Figure 12.** Deformation Z close to the ground.

From Figure 13, the angle change law is basically the same as the *Y*-axis loading situation. The element rotation angle of the LM model is kept approximately constant. The element rotation angle of the FEM has an oscillating process, and finally approaches the same value. The element rotation angle of the ANCF method changes smoothly, and finally converges to a slightly larger value.

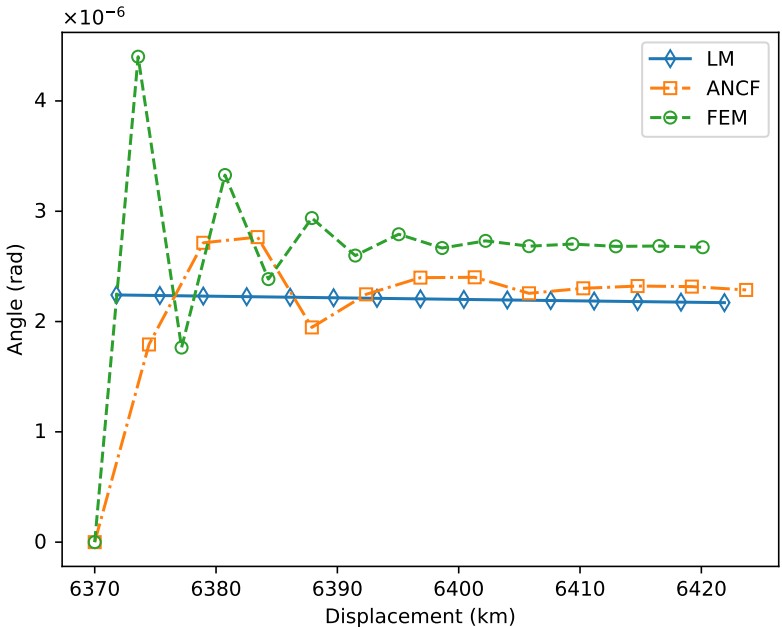

**Figure 13.** Nodal angle $\phi$ close to the ground.

### 4.3. Computational Efficiency

Considering the computational efficiency of each method, the calculated residual of each step is divided by the residual of the first step to obtain the relative value of the residual. Its relationship with the number of calculation iterations is shown in Figure 14:

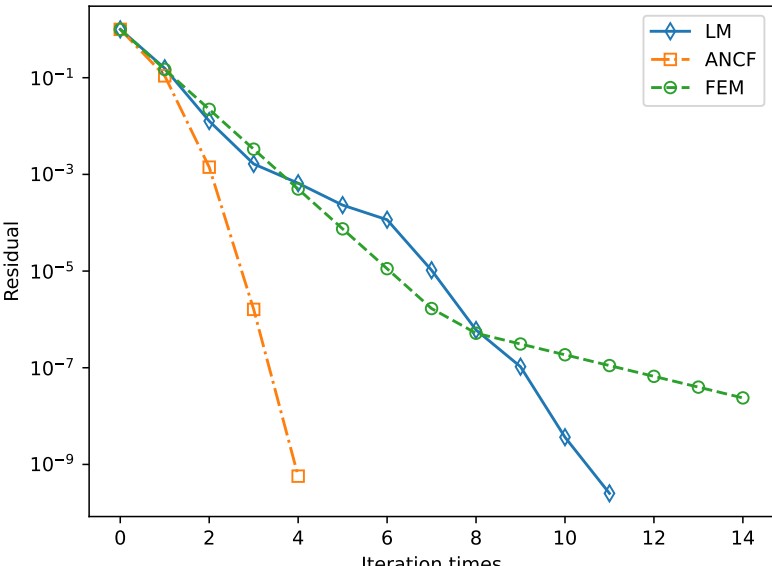

**Figure 14.** Relative value of the calculation residual.

It can be found that the ANCF method has the fastest convergence speed and a trend of accelerating convergence, followed by LM, and the FEM is the slowest.

As the number of iterations increases, the convergence speed of the FEM gradually slows. This is because the stiffness matrix of the system in the FEM is a constant matrix, and each iteration is updated by changing the coordinates to affect the structural force of the node. When the space coordinate change of the rope decreases, the change of the structural force will not converge quickly. However, due to the cumulative effect of the length of the rope, the small changes in each rope segment will accumulate to the change in the position of the zenith anchor, which causes the speed of convergence to decrease. This phenomenon is especially obvious when loading horizontally.

When the LM model achieves better accuracy, the number of elements reach a certain value, which will decrease calculation efficiency. However, since the stiffness matrix of the nonlinear equation system of the LM model is not a constant matrix, the convergence speed will be faster than that of the FEM.

The ANCF method can use a smaller number of elements to achieve the same accuracy due to the higher order of its own elements. There are also nonlinear equations with nonlinear stiffness, although it brings some difficulty to the solution, but the convergence speed is very fast.

## 5. Discussion

### 5.1. Axial Loading

The calculation results of FEM and LM model are very close: both methods use true strain as a measure of rope deformation, while the ANCF method uses Green–Lagrangian strain as a measure of rope deformation. The difference in deformation measures will result in a numerical difference in the stress and strain obtained under the same deformation. In the calculation of the space elevator system rope, the displacement of the zenith anchor is the sum of the deformation of each point of the rope; thus, these numerical differences will amount to the difference in the calculation of stress and displacement. In actual application, this should be compared with the actual situation and a more accurate strain measurement should be used.

### 5.2. Lateral Loading

In the case of lateral loading, the calculation results of the three methods are basically the same; however, since the bending stiffness of the rope is not involved in the LM model,

it cannot be obtained when the flexible deformation contour line is locally required. The beam element of FEM uses an approximate curvature, which causes the overall system stiffness matrix in FEM to remain unchanged. This feature can turn the system's nonlinear equations into a linear equation, while the gravitational and centrifugal force received by each node will change as the coordinates of the node change. Therefore, it is necessary to continuously update the loaded nodal force in the calculation. Under certain conditions, the result of the calculation will not converge.

For example, when $m_c$ is $1.96 \times 10^6$ kg, lateral force is $1 \times 10^5$ N, the result of the calculation is shown in the Figure 15.

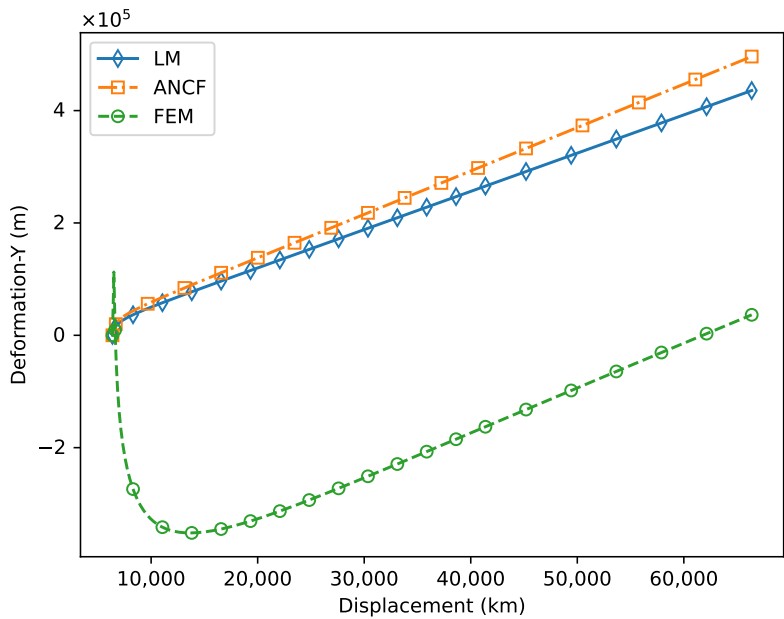

**Figure 15.** Example case of FEM failure.

In this case, the results of ANCF method and LM model calculations are basically the same, but the FEM calculation failed. The mass of the zenith anchor is reduced, resulting in a drop in the rope tension. When loading laterally, the change in the nodal force caused by the displacement of the previous step leads to a greater nodal displacement, which leads to non-convergence of the calculation results.

### 5.3. Computational Efficiency

The ANCF method and LM model use Newton's method for iterative solutions. Although the Jacobian matrix needs to be solved numerically, the calculation cost is relatively high but the total number of calculation steps is reduced, thereby reducing the calculation time.

FEM obtains the result through the solution of multi-step linear equations, but when the displacement change is close to the force change, the speed of convergence will be very slow or the convergence will fail. The stability is worse than that of the ANCF method and the LM model.

### 6. Conclusions

In this paper, ANCF method were applied to the calculation of the space elevator system, to solve the problem of insufficient description of the rope flexibility of the space elevator system. The results show that:

In the case of axial loading, the calculation results of the ANCF method are not very different from the results of the FEM and LM models.

In the case of lateral loading, the calculation results of the ANCF method are basically the same as the results of the FEM and LM models, but they can better reflect the local flexible deformation of the ladder rope, and have a better calculation stability than FEM.

Under the same calculation accuracy, the ANCF method can use fewer elements, and the speed of convergence is faster than the FEM and LM models.

**Author Contributions:** Conceptualization, S.L.; methodology, S.L. and Y.F.; software, S.L.; validation, S.L.; formal analysis, S.L.; investigation, Y.F.; resources, Y.F.; data curation, S.L.; writing—original draft preparation, S.L.; writing—review and editing, S.L.; visualization, S.L.; supervision, N.C.; project administration, Y.F.; funding acquisition, Y.F. All authors have read and agreed to the published version of the manuscript.

**Funding:** This research received no external funding.

**Institutional Review Board Statement:** Not applicable.

**Informed Consent Statement:** Not applicable.

**Data Availability Statement:** Not applicable.

**Conflicts of Interest:** The authors declare no conflict of interest.

## Abbreviations

The following abbreviations are used in this manuscript:

| | |
|---|---|
| LM | Lumped mass |
| FEM | Finite element method |
| ANCF | Absolute nodal coordinate formulation |

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
