# Peer review of "Application of Absolute Nodal Coordinate Formulation in Calculation of Space Elevator System"

_applsci, doi:10.3390/app112311576_

Round 1
Reviewer 1 Report
This paper considers the problem of modelling the static deformation of a space elevator. The authors propose to use a relatively advanced modelling framework (ANCF) which has been shown to be relatively accurate at modelling high-deformation structures. The paper is organized relatively well and the results are interesting, in that they diverge from the standard finite-element and lumped-mass models which are more commonly used. However, the paper is relatively poorly written, so it is difficult to understand how the method works -- at least at sufficient detail to verify the results are correct. Because the numerical analysis diverges from the LM and FEM models, this is particularly important -- it should be absolutely clear that the proposed results are correct and that previous methods are wrong. For this reason, I recommend the authors submit a revised manuscript which more carefully details the method and, if possible, verifies the numerical results for some case where the analytic solution is known.
Notes:
\begin{itemize}
\item The introduction is relatively well-organized.
\item Page 2: "countless rigid rods" - what does this mean? uncountable?
\item page 2: "applid"
\item The description and labelling of Fig. 2 is insufficient.
\item The definition of coordinate system is insufficient. What is the x-direction? Is it rotating?
\item Page 3: "this paper established" -- which paper?
\item page 4: t is missing in the RHS of eqn 2
\item page 5: "but the centrifugal force not"
\item Page 5: It is claimed that (17) can be solved by newton iteration. First, it is not clear what $Q_e$ is on the LHS of (17). Next, Newton iteration requires analytic formulation of the derivatives of the equation to be solved and is notoriously unstable, so the authors need to explain very carefully how Newton iteration can be used to solve this equation: what are the variables, howe are the derivatives computed, how it the algorithm initialized, what is the domain of convergence, etc.
\item More explanation is needed for the figures.
\item page 11: "which destined that the calculation" -- meaning unclear.
\item There are many points in this paper which are not well-explained and/or in good english.
\end{itemize}
Reviewer 2 Report
REVIEW
on article
Application of absolute nodal coordinate formulation in calculation of space elevator system
Shihao Luo, Youhua Fan, and Naigang Cui
SUMMARY
The article is devoted to the application of the formulation of absolute nodal coordinates when calculating the system of space elevators.
The authors identified the object of research: a potentially new space transport system for transporting payloads into outer space. Existing calculation models cannot fully reflect the main mechanical characteristics of a flexible rope, which is a part of the system, and cannot accurately describe significant deformations during the movement of a flexible rope. To ensure the accuracy of existing models, a large number of elements are required, so simulation is ineffective.
The authors propose a solution to these problems by using the ANCF method to calculate the space elevator system. They compared the distribution of axial stresses, deformation in each direction, and calculation efficiency of the three methods.
The authors' results show that the ANCF method can use fewer elements to achieve the same calculation accuracy, and it describes well the flexibility of the ladder rope system. The convergence rate is also the fastest of the three methods.
The article is interesting from the point of view of applied engineering science.
The authors' research deserves publication in the journal, but first, it is necessary to correct the comments below.
COMMENTS
- The authors must redo the Abstract and bring it in compliance with the requirements of the Applied Sciences journal. The scientific problem is not described (Background). The scientific novelty is not indicated. The authors presented an excellent engineering problem, however, did not highlight the scientific novelty. Editors strongly encourage authors to use the following style of structured abstracts, but without headings: (1) Background: Place the question addressed in a broad context and highlight the purpose of the study; (2) Methods: Describe briefly the main methods or treatments applied; (3) Results: Summarize the article's main findings; and (4) Conclusions: Indicate the main conclusions or interpretations. The abstract should be an objective representation of the article.
- The Abstract contains less than 200 words. It should be expanded to the journal's requirement of 200 words.
- There is a mistake in the Abstract on line 7 and onwards. It is necessary to correct the applid to applied
- There are 2 affiliations in the list of authors, and 3 in the list of affiliated universities.
- In the presented section "Introduction", the literature review contains a small number of literature sources published over the past 5 years. The survey should be expanded by adding 10-15 research-related sources published from 2017 to 2021.
- The literature review contains four references 14-17 to the same author, thus artificially increasing self-citation. Perhaps 1-2 references will suffice to reflect the direction and results of the author's research.
- At the end of the "Introduction" section, a problem is formulated that the authors are solving, but it is also necessary to highlight the purpose, objectives, and scientific novelty of the research.
- It is not clear from the problem statement how the authors take into account the variable properties of gravity with distance from the surface.
- In Figure 1 there is no designation L shown in front of the figure. Also, in the description of the designations for Figure 1, there is no description of what Oo is. It is necessary to clarify.
- How the model takes into account the own weight of the beam taking into account the change in the acceleration of gravity?
- How are the speeds of rotation of the Earth and the masses M in orbit synchronized?
- The problem statement considers only the axial loading of the beam. How is the possible rotation of the mass M about the axis of the beam taken into account?
- A more detailed description of Figure 2 should be added: describe what is depicted, all the symbols.
- Figure 3 shows the LM model, but there is no detailed description of it, the interpretation of Figure 3 is not presented. It is necessary to add.
- Figures 10 and 11 are not sufficiently described. You need to add 1-2 paragraphs.
- Taking into account all the nonlinear properties of the problem, why is the cross-sectional area of the beam constant A=0.01 m2?
- Section 5 "Discussion and Conclusions" should be divided into two sections "Discussion" and "Conclusion". In the "Discussion" section, it is necessary to provide a detailed justified comparison of the results obtained in the presented study with the works of other authors who have studied similar topics.
- In the section "Conclusions" it is necessary to reflect what is the novelty of the results obtained and what are the prospects for their practical application. It is necessary to revise this section, remove from it the interpretation of the results and discussion, list the main results obtained and what problems were solved.
The article is very controversial. It is unclear what parameters of the beam and mass in orbit were obtained as a result of solving the problem. The kinematic characteristics must be supplemented with dynamic ones.
Round 2
Reviewer 1 Report
The authors have made a serious effort to address my concerns from the previous review and as a result the manuscript is much improved. For final feedback, I would consider adding more explanations to the Figures. Specifically, for Figure 15, where FEM fails, make sure the caption indicates this is a failure case for FEM and what caused it. Otherwise, one has to hunt through the text to figure out what is going on.
Author Response
We sincerely thank the reviewer for thoroughly examining our manuscript and providing very helpful comments to guide our revision.
The caption of Figure 15 has been revised.
We would like to thank the referee again for taking the time to review our manuscript.
Reviewer 2 Report
All my comments are taken into account. The article has been corrected.
The authors submitted a controversial article. The concept presented in the article is controversial. However, perhaps in the distant future, this idea will be implemented and the preliminary developments of the authors will be used.
I wish the authors success!
Author Response
We sincerely thank the reviewer for thoroughly examining our manuscript and providing very helpful comments to guide our revision.